# Tuberculin skin test and Interferon-gamma release assay agreement, and associated factors with latent tuberculosis infection, in medical and nursing students in Bandung, Indonesia

Lika Apriani [1,2,3]*, Susan McAllister[3], Katrina Sharples[3,4], Isni Nurul Aini[1], Hanifah Nurhasanah[1], Dwi Febni Ratnaningsih[1], Agnes Rengga Indrati [1,5], Rovina Ruslami[1,6], Bachti Alisjahbana[1,7], Reinout van Crevel[8], Philip C. Hill[3]

1 TB Working Group, Research Center for Care and Control of Infectious Disease, Universitas Padjadjaran, Bandung, Indonesia, 2 Department of Public Health, Faculty of Medicine, Universitas Padjadjaran, Bandung, Indonesia, 3 Centre for International Health, Division of Health Sciences, University of Otago, Dunedin, New Zealand, 4 Department of Mathematics and Statistics, University of Otago, Dunedin, New Zealand, 5 Department of Clinical Pathology, Faculty of Medicine, Universitas Padjadjaran, Bandung, Indonesia, 6 Department of Biomedical Sciences, Faculty of Medicine, Universitas Padjadjaran, Bandung, Indonesia, 7 Department of Internal Medicine, Faculty of Medicine, Universitas Padjadjaran, Bandung, Indonesia, 8 Department of Internal Medicine, Radboud University Medical Center, Nijmegen, The Netherlands

* lika.apriani@unpad.ac.id

## Abstract

### Background

No gold standard diagnostic test exists for latent tuberculosis infection (LTBI). The intra-dermal tuberculin skin test (TST) has known limitations and Interferon-gamma release assays (IGRA) have been developed as an alternative. We aimed to assess agreement between IGRA and TST, and risk factors for test positivity, in Indonesian healthcare students.

### Methods

Medical and nursing students starting their clinical training were screened using IGRA and TST. Agreement between the two tests was measured using Cohen's Kappa coefficient. Logistic regression was used to identify factors associated with test positivity.

### Results

Of 266 students, 43 (16.2%) were IGRA positive and 85 (31.9%) TST positive. Agreement between the two tests was 74.7% (kappa 0.33, 95% CI 0.21–0.45, P<0.0001). Students who had direct contact with family or friends with TB were less likely to be test positive using IGRA (AOR 0.18, 95% CI 0.05–0.64) and using TST (AOR 0.51, 95% CI 0.26–0.99).

### Conclusion

Test positivity for LTBI was lower when measured by IGRA than by TST, with poor agreement between the two tests. Known close TB contact was unexpectedly negatively

**Data Availability Statement:** All relevant data are within the manuscript and its Supporting information files.

**Funding:** LA was supported by the University of Otago Doctoral Scholarship. Research cost was sponsored by the Centre for International Health with donations from Mercy Hospital Charitable Trust Dunedin, New Zealand. Qiagen provided IGRA diagnostic kits. Publication cost was supported by Universitas Padjadjaran. The funder had no role in study design, data collection and analysis, decision to publish, or manuscript preparation.

**Competing interests:** NO authors have competing interests.

associated with positivity by either test. Longitudinal studies may be required to help determine the best test for LTBI in healthcare students in Indonesia.

## Introduction

In high tuberculosis (TB) endemic countries such as Indonesia, identifying high-risk sub-population with latent TB infection (LTBI) is effective for TB control. The provision of preventive treatment for those with LTBI can prevent individuals from developing TB disease [1, 2]. Medical and nursing students are vulnerable to *Mycobacterium tuberculosis* exposure and infection during their clinical training at healthcare facilities and are therefore considered to be a high-risk group [2]. Screening for TB disease and LTBI should therefore be considered prior to commencing clinical training [2, 3], however, there is no gold standard diagnostic test for identifying LTBI.

There are two accepted but imperfect tests for LTBI: tuberculin skin test (TST) and Interferon-gamma release assays (IGRA). TST consists of an intradermal injection of 5 tuberculin units (TU) of purified protein derivative (PPD)-S (Siebert) or 2 TU PPD research tuberculin (RT) 23 and is performed using the Mantoux technique [4]. These two types of TU are considered equivalent [5]. Even though TST is widely used and inexpensive, it has several known limitations in specificity, especially in the presence of prior Bacille Calmette-Guerin (BCG) vaccination [6, 7]. A further limitation with the use of TST is that there are logistical issues of requiring a return visit by a specialist nurse or doctor after two to three days to read the induration size. Self-reading is possible but has been associated with a high error rate [8]. Lastly, there is a global shortage of TST which is attributed to market forces [9].

IGRAs are blood tests that measure the secretion of interferon-gamma (IFN-γ) by T-cell after stimulation by antigens that are fairly specific to the *M. tuberculosis* complex (with the exception of BCG sub-strains) [8]. Therefore, their specificity for *M. tuberculosis* is higher than with TST. IGRAs are not subject to sensitisation or boosting and they require only one visit to a healthcare facility [10]. There are two available commercial IGRA tests: The (QuantiFER-ON-TB Gold Plus (QFT-Plus)) assay (the fourth generation QFT assay produced by Cellestis/Qiagen, Carnegie, Australia) and the T-SPOT.TB assay (Oxford Immunotec, Abingdon, United Kingdom). Limitations on the use of IGRA are that it is more expensive, requires venepuncture of the patient, and requires specific laboratory expertise [9, 11].

Previous reviews of healthcare workers (HCWs) in low and middle-income countries [3] and in countries with a low and intermediate TB incidence [10] reported a lower LTBI prevalence estimate when measured by IGRA than by TST. Poor agreement between TST and IGRA has been reported in two systematic reviews [12, 13] which reported a kappa statistic of 0.28 (95% CI 0.22–0.35) [12], and an overall kappa statistic of 0.27 (95% CI 0.22–0.32), although the agreement was higher in high-burden TB countries than in low-burden countries [13].

Indonesia, has the the second highest TB caseload in the world with approximately 1,060,000 cases annually and an incidence rate of 385 per 100,000 population in 2022 [14]. To our knowledge, only one study has compared IGRA and TST in HCWs in Indonesia and this showed poor agreement between the two tests (kappa statistic of 0.34) [15]. Little is, however, known about TST and IGRA agreement in diagnosing LTBI in medical and nursing students before being exposed to the healthcare setting. We therefore undertook a study to assess IGRA and TST agreement as screening tests for LTBI in medical and nursing students who started their clinical program in a tertiary hospital in Bandung, Indonesia, and investigated factors associated with LTBI.

## Materials and methods

### Study design and procedures

The study was part of a cohort study of medical and nursing students at Universitas Padjadjaran who entered their clinical training in Hasan Sadikin General Hospital in 2017 [16]. This cross-sectional study was conducted only in those entering the training in January 2017 who were tested for both TST and IGRA. All eligible students provided written consent. We obtained ethical approval from the University of Otago Ethics Committees, New Zealand (HE16/005) and the Universitas Padjadjaran Ethics Committees Bandung, Indonesia (No 820/UN6.C1.3.2/ KEPK/PN/2016).

All consenting students were interviewed for possible TB risk factors and tested by IGRA and TST. A questionnaire was used to collect data on demographic characteristics (age, gender, student type, housing, and ethnicity), clinical characteristics (history of BCG vaccination and/or evidence of a BCG scar, human immunodeficiency virus (HIV) status, whether they had any immunocompromised condition), behavioral characteristics (smoking, alcohol consumption), and TB exposure at a health facility or in the home. Students who had a positive prior TST and/or IGRA or a history of TB disease prior to study enrolment were ineligible.

To avoid any potential boosting effect of TST on the IGRA result, sampling for IGRA was performed just prior to TST. The commercial IGRA QFT-Plus test was used and interpreted as per the manufacturer's instructions [17]. Blood samples were collected into four tubes (Nil, TB1, TB2, and Mitogen), incubated at 37°C for 24 hours, centrifuged and stored at 4°C. The enzyme-linked immune sorbent assay (ELISA) was performed manually in batches. TST was performed using 0.1 ml (2 international units) tuberculin of purified protein derivative (PPD RT23 Biofarma® Bandung) and administered by an experienced study nurse. It was given on the volar side of the forearm and the skin reaction read by the same study nurse between 48 and 72 hours after TST placement. An induration size of ≥10 mm was considered positive [18].

Participants were provided with their IGRA and TST results in a sealed envelope in person. Those with a positive result were asked for a review to reassess for symptoms of TB disease, referral for diagnostic evaluation for TB disease, and appropriate treatment. A small number of participants were expected to have TB disease (approx. 1%). In keeping with national guidelines when implementing this study, no LTBI treatment was offered for HCWs or students with a positive result. LTBI treatment at the time was recommended only for children under 5 years of age who are household contacts of people with TB, and people living with HIV [19].

### Data management and statistical analysis

All data were entered into a database created in RedCap® [20]. Data were transferred into Stata® version 14.2 (Stata Corp LP, Texas, US) for analysis. Descriptive statistics and the median (IQR) for age and body mass index (BMI) were calculated. Both IGRA and TST were recorded as positive or negative. Agreement between the IGRA and TST results was assessed using the kappa statistic [21]. Interpretation of Cohen's kappa as suggested by McHugh [22] was as follows: values 0.00–0.20 as none, 0.21–0.39 as minimal, 0.40–0.59 as weak, 0.60–0.79 as moderate, and 0.80–0.90 as strong, and above 0.90 as almost perfect agreement. In addition, Pearson's correlation was also calculated using the quantitative results of IGRA (TB1-Nil) and TST induration (in mm). The relationship between IGRA and TST positivity and the measures of exposure variables were examined using univariate logistic regression and estimates are presented as odds ratios (OR). Estimates adjusted for characteristics and exposures were obtained using modified backward-stepwise regression and are presented as adjusted OR (AOR). Age,

gender, student type, and BCG vaccination were retained in the multivariate model. Other variables were also retained in the model if the p-values was less than 0.2. Continuous variables were retained in their linear forms because complex fractional polynomials did not improve model fit for any of them. All the reported p-values were two-sided, and p-values less than 0.05 were considered statistically significant.

## Results

Of the 301 students starting clinical training in January 2017, 291 agreed to be considered for the study, and 266 were eligible (Fig 1). Reasons for non-eligibility were history of TB (n = 24) and prior TST positivity (n = 1). The majority of the students included were medical students (89.1%). The median age was 22 years, 73.3% were female, and most (80.1%) were living in student housing. A third was Sundanese (the main ethnicity in West Java Province). Only a small proportion of students had ever smoked (9.8%) or ever consumed alcohol (8.3%) (Table 1). The majority of students (88.7%) had either a history of BCG vaccination or proof of a BCG scar. Of the students, none reported having HIV, and 7 (2.6%) were receiving steroid medication. BMI was 22.3 kg/m2 on average. Owing to the nature of the curriculum, 80.1% of students had completed some training in a healthcare facility before enrolling in the study. Just over half (52.3%) had some direct contact with a TB patient and a quarter had direct contact with family or friends who had been diagnosed with TB (25.9%).

Of the 266 students, IGRA was positive in 43 (16.2%, 95% CI 12.0–21.2), and TST was positive in 85 (31.9%, 95% CI 26.4–37.9) (Table 2). None of the students with positive IGRA and/or TST had TB disease at the time of examination. One student who had an indeterminate IGRA result and positive TST result was not included in the agreement and risk factor analysis.

Table 3 shows the concordance and discordance between IGRA and TST positivity, and kappa statistic. Agreement between the two tests was 74.7% (kappa 0.33, 95% CI 0.21–0.45, P<0.0001). A scatter plot shows a moderate positive correlation between the levels of IGRA and TST induration (r = 0.46, p<0.001) (Fig 2).

The only factor found to be significantly associated with IGRA positivity was in the multivariate analysis, where students who had direct contact with family or friends with TB had a lower risk (AOR 0.18, 95% CI 0.05–0.64). Students who had been involved in another training program before study enrolment and who had direct contact with family or friends with TB had a lower risk of being TST positive in the univariable analysis (OR 0.54, 95% CI 0.29–0.99; and OR 0.50, 95% CI 0.26–0.96, respectively). However, after adjustment, only direct contact with family or friends with TB stayed significant (AOR 0.51, 95% CI 0.26–0.99) (Table 4).

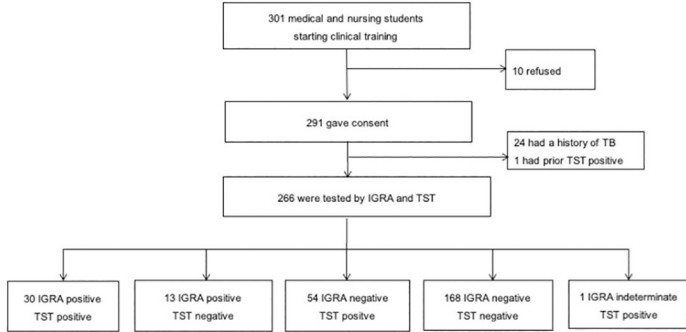

**Fig 1. Flow diagram of subject recruitment.**

**Table 1. Demographic, clinical characteristics, and prior TB exposure in medical and nursing students (n = 266).**

| Characteristics | | Students n (%) |
|---|---|---|
| **Demographic characteristics** | | |
| Student type | Medical | 237 (89.1) |
| | Nursing | 29 (10.9) |
| Age (years) median, IQR | | 21.7 (21.3–22.4) |
| Gender | Female | 195 (73.3) |
| Housing | Family housing | 53 (19.9) |
| | Student housing | 213 (80.1) |
| Ethnicity | Sundanese | 94 (35.3) |
| | Javanese | 61 (22.9) |
| | Minangnese | 48 (18.0) |
| | Other | 63 (23.7) |
| Smoking | Current or ex-smoker | 26 (9.8) |
| Alcohol consumption | Yes | 22 (8.3) |
| **Clinical characteristics** | | |
| BCG vaccination | Yes | 236 (88.7) |
| Diagnosed with HIV | Yes | 0 (0.0) |
| Any other immunocompromised condition (steroid therapy, etc) | Yes | 7 (2.6) |
| Body mass index (kg/m2) median, IQR | | 22.3 (20.1–25.7) |
| Body mass index (kg/m2) | Underweight (<18.5) | 31 (11.7) |
| | Normal (18.5–24.9) | 165 (62.0) |
| | Overweight (25.0–29.9) | 58 (21.8) |
| | Obese (>30) | 12 (4.5) |
| **TB exposure prior to study enrolment** | | |
| Involved in other training in health care facility | Yes | 213 (80.1) |
| Any direct contact with TB patient or participation in sputum collection or examination | Yes | 139 (52.3) |
| Any direct contact with family or friends who have been diagnosed with TB | Yes | 69 (25.9) |

BCG = bacille calmette-guerin, BMI = body mass index, HIV = human immunodefiency virus, IQR = inter quartile range, TB = tuberculosis

## Discussion

In this study, test positivity for LTBI was lower when measured by IGRA than by TST, with minimal agreement between the two tests. Using either IGRA or TST, students who had direct contact with family or friends were unexpectedly negatively associated with test positivity.

**Table 2. Interferon-gamma release assay and tuberculin skin test positivity (n = 266).**

| Definition | Total (n) | Positive (n) | % | 95% CI |
|---|---|---|---|---|
| **IGRA** | | | | |
| Medical and nursing students | 266 | 43 | 16.2 | 12.0–21.2 |
| Medical students | 237 | 40 | 16.9 | 12.3–22.3 |
| Nursing students | 29 | 3 | 10.3 | 2.2–27.4 |
| **TST** | | | | |
| Medical and nursing students | 266 | 85 | 31.9 | 26.4–37.9 |
| Medical students | 237 | 75 | 31.6 | 25.8–38.0 |
| Nursing students | 29 | 10 | 34.5 | 17.9–54.3 |

CI = confidence interval, IGRA = interferon-gamma release assay, TST = tuberculin skin test

**Table 3. Kappa statistic for agreement between IGRA and TST positivity (n = 265).**

| IGRA | TST | | Total | Agreement | Kappa | 95% CI | P value |
|---|---|---|---|---|---|---|---|
| | Positive | Negative | | | | | |
| Positive | 30 | 13 | 43 | 74.7 | 0.33 | 0.21–0.45 | <0.0001 |
| Negative | 54 | 168 | 222 | | | | |
| Total | 84 | 181 | 265 | | | | |

IGRA = interferon-gamma release assay, TST = tuberculin skin test

IGRA positivity in our study was lower than TST positivity, consistent with that reported in previous reviews [3, 10] and studies in South Africa [23] and in Ethiopia [24]. While it's different from a study in India [25], which showed IGRA positivity was slightly higher than TST positivity. This discrepancy is probably caused by the underlying country TB prevalence, clinical circumstances, variations in exposure to non-tuberculous mycobacteria which can confound the TST results, and varying TB infection control practices in healthcare facilities, in addition to the study participants's demographic characteristics, and recruiting time.

The low agreement between the two tests was consistent with two earlier reviews between IGRA and TST in 29 HCW studies (kappa 0.28, 95% CI 0.22–0.35) by Lamberti et al. [12], and in 30 HCW studies (kappa 0.27, 95% CI 0.22–0.32) by Doosti-Irani et al [13]. The agreement in our study was also similar to the only known HCW study in Indonesia that reported a kappa statistic of 0.34 comparing IGRA in TST [15]. However, positivity in that study was higher with IGRA than with TST which is contrary to that reported in other reviews [3, 10].

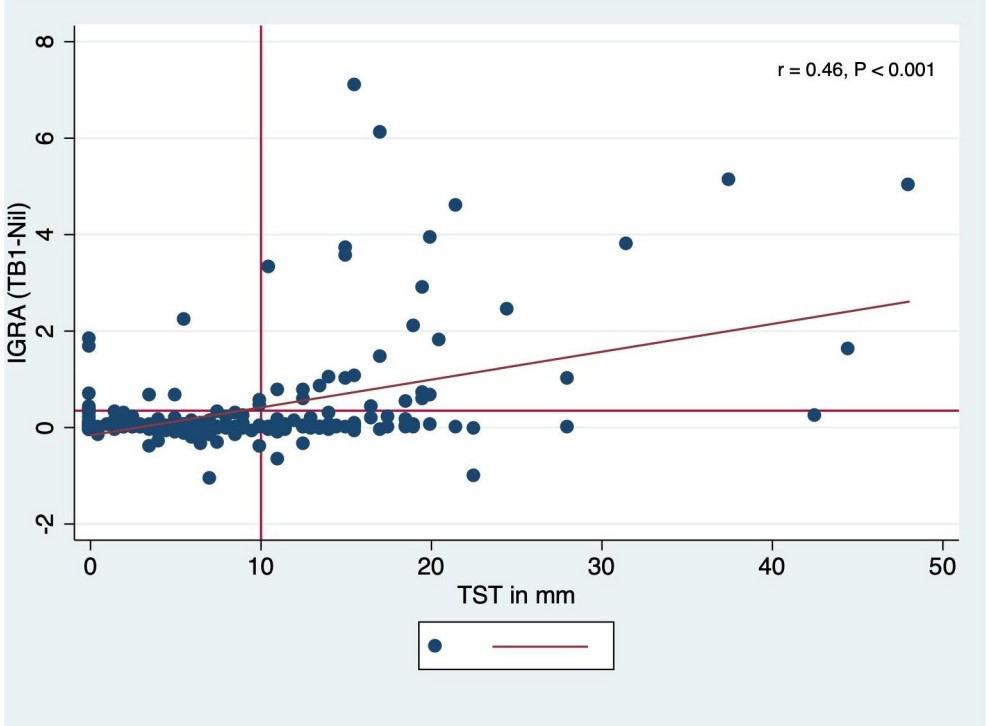

**Fig 2. Scatter graphs showing correlations between quantitative IGRA and TST test results.**

**Table 4. Association between demographic characteristics, medical history, TB exposure, and IGRA and TST positivity in medical and nursing students.** Results of univariate and multivariate logistic regression (IGRA = 266, TST = 265).

| Variable | | Positive IGRA | Negative IGRA | OR (95% CI) | AOR (95% CI) | Positive TST | Negative TST | OR (95% CI) | AOR (95% CI) |
|---|---|---|---|---|---|---|---|---|---|
| | | n (%) | n (%) | | | n (%) | n (%) | | |
| Total students | - | 43 (16.2) | 222 (83.8) | - | - | 85 (31.9) | 181 (68.1) | - | - |
| Age, median (years) | - | 21.8 (21.5–22.7) | 21.7 (21.2–22.4) | 1.14 (0.86–1.51) | 1.10 (0.82–1.47) | 21.8 (21.5–22.6) | 21.7 (21.3–22.3) | 1.19 (0.94–1.49) | 1.13 (0.88–1.44) |
| Gender | Male | 9 (12.7) | 62 (87.3) | 1 | 1 | 22 (31.0) | 49 (69.0) | 1 | 1 |
| | Female | 34 (17.5) | 160 (82.5) | 1.46 (0.66–3.22) | 1.73 (0.76–3.93) | 63 (32.3) | 132 (67.7) | 1.06 (0.59–1.91) | 1.15 (0.63–2.10) |
| Student type | Nursing | 3 (10.3) | 26 (89.7) | 1 | 1 | 10 (34.5) | 19 (65.5) | 1 | 1 |
| | Medical | 40 (16.9) | 196 (83.1) | 1.77 (0.51–6.12) | 1.41 (0.37–5.44) | 75 (31.7) | 162 (68.3) | 0.88 (0.39–1.98) | 0.77 (0.32–1.89) |
| Housing | Student housing | 36 (16.9) | 177 (83.1) | 1 | | 68 (31.9) | 145 (68.1) | 1 | |
| | Family housing | 7 (13.5) | 45 (86.5) | 0.76 (0.32–1.83) | - | 17 (32.1) | 36 (67.9) | 1.01 (0.53–1.92) | - |
| Ethnicity | Other | 30 (17.4) | 142 (82.6) | 1 | | 60 (34.9) | 112 (65.1) | 1 | |
| | Sundanese | 13 (13.9) | 80 (86.1) | 0.77 (0.38–1.56) | - | 25 (26.6) | 69 (73.4) | 0.68 (0.39–1.28) | - |
| BCG vaccination | No | 6 (20.0) | 24 (80.0) | 1 | 1 | 11 (36.7) | 19 (63.3) | 1 | 1 |
| | Yes | 37 (15.7) | 198 (84.3) | 0.75 (0.29–1.95) | 0.61 (0.22–1.70) | 74 (31.4) | 162 (68.6) | 0.79 (0.36–1.74) | 0.73 (0.32–1.66) |
| Smoking | Never | 41 (17.2) | 198 (82.8) | 1 | | 75 (31.3) | 165 (68.8) | 1 | |
| | Current/ex-smoker | 2 (7.7) | 24 (92.3) | 0.45 (0.09–1.77) | - | 10 (38.5) | 16 (61.5) | 1.38 (0.60–3.17) | - |
| Alcohol consumption | No | 40 (16.5) | 203 (83.5) | 1 | | 77 (31.6) | 167 (68.4) | 1 | |
| | Yes | 3 (13.6) | 19 (86.4) | 0.80 (0.22–2.84) | - | 8 (36.4) | 14 (63.6) | 1.24 (0.50–3.08) | - |
| BMI (kg/m2) median, IQR | - | 22.4 (20.4–26.2) | 22.2 (20.1–25.6) | 1.01 (0.93–1.09) | - | 22.5 (20.1–26.2) | 22.1 (20.2–24.9) | 1.03 (0.97–1.09) | - |
| Involved in other programme in health care facility prior to recruitment | No | 13 (24.5) | 40 (75.5) | 1 | 1 | 23 (43.4) | 30 (56.6) | 1 | 1 |
| | Yes | 30 (14.2) | 182 (85.8) | 0.51 (0.24–1.06) | 0.55 (0.26–1.23) | 62 (29.1) | 151 (70.9) | **0.54 (0.29–0.99)** | 0.58 (0.31–1.10) |
| Direct contact with TB patient or participation in sputum collection/ examination | No | 22 (17.5) | 104 (82.5) | 1 | | 44 (34.7) | 83 (65.3) | 1 | |
| | Yes | 21 (15.1) | 118 (84.9) | 0.84 (0.44–1.62) | - | 41 (29.5) | 98 (70.5) | 0.79 (0.47–1.32) | - |
| Direct contact with family or friends with TB | No | 40 (20.4) | 156 (79.6) | 1 | 1 | 70 (35.5) | 127 (64.5) | 1 | 1 |
| | Yes | 3 (4.4) | 66 (95.6) | 0.18 (0.05–0.59) | **0.18 (0.05–0.64)** | 15 (21.7) | 54 (78.3) | **0.50 (0.26–0.96)** | **0.51 (0.26–0.99)** |

AOR = adjusted odds ratio, BCG = Bacille Calmette-Guerin, BMI = body mass index, CI = confidence interval, IGRA = interferon-gamma release assay, IQR = inter quartile range, OR = odds ratio

Percentages are by row

In our study, unexpectedly, direct contact with family or friends with TB was negatively associated with both positive IGRA and TST. This is contrary to what would be expected, and the reasons are not entirely clear. One possible contributor could be that these students were more aware of TB disease, its symptoms and means of prevention before entering the clinical training programme, providing some degree of protection. This has been seen in a previously published HCW study in a tertiary hospital where HCWs in high and very high-risk areas and

procedures were more aware of TB disease; when providing care for their patients, they wore respirator masks the majority of the time, although, mask-wearing was uncommon in other parts of the hospital [26].

Our study adds to the large amount of evidence on the level of agreement between two available tests for LTBI. It is the first study conducted in medical and nursing students in Indonesia before commencing their clinical training programme. The main limitation, however, is the inability to truly guarantee the accuracy of both tests due to the absence of a gold standard test. Even though both TST and IGRA are acceptable and widely used, they are both imperfect and have several limitations, as described earlier.

This study has some implications for practice. The difference in positivity rates for IGRA and TST in our study may create uncertainty regarding the diagnosis of LTBI. A review by Salgame et al. [27] recommended that in cases where TST and IGRA are in disagreement, individuals are usually considered to have LTBI unless there is a strong suspicion of a false positive result (such as in repeated BCG vaccination and known highly non-tuberculosis mycobacterial infection in the population). Hence, when using both tests for LTBI screening, if either test is positive, preventive treatment should be considered after undertaking a clinical examination [27]. Providing LTBI screening for HCWs and healthcare students (HCSs) should be more actively considered in Indonesia. Either TST or IGRA could be used, particularly as there is no diagnostic gold standard for LTBI, and both tests are included in the new recommendation by World Health Organization (WHO) [1].

With the current global, and also national shortage of tuberculin in Indonesia, there is some potential to substitute IGRA for TST. However, there is increased cost associated with using IGRA and it requires venapuncture and specific laboratory services, so it may not be feasible in all settings. Therefore, the program manager or clinician could choose the available and affordable test in their facility. Given the high LTBI positivity in our study and the risk of exposure to TB in the clinical setting and acknowledging Indonesia as a resource-limited and high TB burden country, we recommend that the National TB Programme of Indonesia prioritise the allocation of TST usage in all TB care centres, until there is proof of a substantially improved test. Priority should be for individuals at high risk of TB exposure–which includes HCWs. LTBI screening of newly employed HCWs, new HCSs, and current HCWs, and regular repeat screens, if necessary, are advisable. As an example, a routine serial test for LTBI is necessary in some surveillance programs for assessing the effectiveness of TB infection control measures in healthcare facilities. Together with LTBI treatment, it can prevent these high-risk individuals from developing TB disease.

## Conclusion

Test positivity for LTBI was lower when measured by IGRA than by TST, with poor agreement between the two tests. Known close TB contact was unexpectedly negatively associated with positivity by either test. Longitudinal studies may be required to help determine the best test for LTBI in healthcare students in Indonesia.

## Supporting information

**S1 Dataset.**
(PDF)

**S1 Questionnaire.**
(PDF)

## Acknowledgments

The authors thank the Dean of Faculty of Medicine, Dean of Faculty of Nursing Universitas Padjadjaran, Director of Hasan Sadikin General Hospital for facilitating the study, and the medical and nursing students for participating in the study. The authors are grateful for the dedicated field, laboratory, and data management team who recruited the cohort including Novianti, Deni, Wiwik Pratiwi, Gita Pujisari, Andini Cahya Nurani, Abdul Kamil, Andhika Candra Guntara, Lailani, Luri Marulitasari, Mila Zahra Latief, Fitria Utami, Yusak Satra Atmaja, Anbarunik Puteri Danthin, Nur Hanifiani, Nopi Susilawati, Runi Rahmawati, Rani Trisnawati, and Deri Pebrilian.

## Author Contributions

**Conceptualization:** Lika Apriani, Susan McAllister, Katrina Sharples, Philip C. Hill.

**Data curation:** Lika Apriani, Isni Nurul Aini, Hanifah Nurhasanah.

**Formal analysis:** Lika Apriani, Katrina Sharples.

**Funding acquisition:** Lika Apriani, Philip C. Hill.

**Investigation:** Lika Apriani, Isni Nurul Aini, Hanifah Nurhasanah, Dwi Febni Ratnaningsih, Agnes Rengga Indrati.

**Methodology:** Lika Apriani, Susan McAllister, Katrina Sharples, Isni Nurul Aini, Hanifah Nurhasanah, Dwi Febni Ratnaningsih, Agnes Rengga Indrati, Philip C. Hill.

**Project administration:** Lika Apriani, Bachti Alisjahbana, Philip C. Hill.

**Resources:** Lika Apriani.

**Supervision:** Lika Apriani, Susan McAllister, Katrina Sharples, Rovina Ruslami, Bachti Alisjahbana, Reinout van Crevel, Philip C. Hill.

**Writing – original draft:** Lika Apriani.

**Writing – review & editing:** Lika Apriani, Susan McAllister, Katrina Sharples, Isni Nurul Aini, Hanifah Nurhasanah, Dwi Febni Ratnaningsih, Agnes Rengga Indrati, Rovina Ruslami, Bachti Alisjahbana, Reinout van Crevel, Philip C. Hill.

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
