## [Decision Letter · Decision Letter 0]

20 Dec 2023

PONE-D-23-29807Tuberculin skin test and Interferon-gamma release assay agreement, and associated factors with latent tuberculosis infection, in medical and nursing students in Bandung, IndonesiaPLOS ONE

Dear Dr. Apriani,

Thank you for submitting your manuscript to PLOS ONE. After careful consideration, we feel that it has merit but does not fully meet PLOS ONE’s publication criteria as it currently stands. Therefore, we invite you to submit a revised version of the manuscript that addresses the points raised during the review process.

We look forward to receiving your revised manuscript.

Kind regards,

Mao-Shui Wang

Academic Editor

PLOS ONE

- https://doi.org/10.1093/trstmh/trab038

In your revision ensure you cite all your sources (including your own works), and quote or rephrase any duplicated text outside the methods section. Further consideration is dependent on these concerns being addressed.

4. In the online submission form, you indicated that [The datasets used and/or analysed during the current study are available from the corresponding author on reasonable request].

Reviewers' comments:

Reviewer's Responses to Questions

**Comments to the Author**

1. Is the manuscript technically sound, and do the data support the conclusions?

Reviewer #1: Yes

Reviewer #2: Yes

2. Has the statistical analysis been performed appropriately and rigorously? 

Reviewer #1: Yes

Reviewer #2: I Don't Know

3. Have the authors made all data underlying the findings in their manuscript fully available?

Reviewer #1: Yes

Reviewer #2: Yes

4. Is the manuscript presented in an intelligible fashion and written in standard English?

Reviewer #1: Yes

Reviewer #2: Yes

5. Review Comments to the Author

Reviewer #1: The manuscript presented interesting findings.

Minor comments:

1. The authors noted that the current study was a part of a preceding study (10.1093/trstmh/trab038). In that earlier study, a total of 379 individuals underwent analysis, with 70 testing positive for IGRA. In the present study, the authors initially identified 301 individuals and subsequently analyzed 266, of whom 43 tested positive for IGRA. Please provide detailed clarification on the inclusion and exclusion processes. A flow diagram illustrating subject recruitment would be beneficial in elucidating the included patients, if deemed necessary.

2. Is the questionnaire available? I would recommend attach it as a supplementary file

3. Line 122-123: Who interpret the TST? Was it the same nurse?

4. Table 1: The majority of the study population came from “ethnicities other than Sundanese”. Can this be specified? as this was the majority (perhaps the top three).

5. Table 4: Please include columns for "negative IGRA" and "negative TST." In the columns "positive IGRA" and "positive TST," what were the denominators used for percentage calculation? For instance, the proportion of males with a positive IGRA was 9 individuals, and the percentage in parentheses was 12.7. How did the authors calculate 12.7% as 9/265 (all included individuals) equal to 3.4%, and 9/43 (positive IGRA only) equal to 20.9%? Were these calculated on a "row" basis? If so, the table should be revised for better clarity.

6. One student with an indeterminate IGRA result (Line 171-172) was not included in the agreement and risk factor analysis. Please specify the TST result.

7. Line 225-26: “The small number of our study subjects could also make the estimate unstable”. What do the authors mean by "unstable"? How many additional samples do the authors anticipate, and what is the rationale for this? I would prefer to delete this sentence instead.

8. (Line 232) the authors are expected to elaborate more on possible NTM in the population as this can affect the false-positive TST (i.e., possible [high?] exposure in the medical/nursing academic environment as well).

Thank you.

Reviewer #2: The study explains the association between TST and IGRA performed on Medical students and nurses before commencement of clinical rotation. I have following minor comments.

1. The study was done almost 6 yrs ago that is Jan-Feb 2017. I do not know the Journal policy it accepts more than 5 yrs old study or not.

2. In introduction, please mention the incidence and prevalence of TB in Indonesia with reference.

3. Line 122. Please write clearly the time for reading, 48 or 72 hrs as it can produce a false negative result.

6. PLOS authors have the option to publish the peer review history of their article (what does this mean?). If published, this will include your full peer review and any attached files.

Reviewer #1: No

Reviewer #2: No

---

## [Author Response · Author response to Decision Letter 0]

1 Feb 2024

We are pleased to resubmit our manuscript entitled “Tuberculin skin test and Interferon-gamma release assay agreement, and associated factors with latent tuberculosis infection, in medical and nursing students in Bandung, Indonesia” for consideration for publication in PLOS ONE.

We would like thank the editor and reviewers for their high quality and constructive review and for careful reading of our manuscript. In this revised version of the manuscript, we have addressed all comments raised by the editor and reviewers. Changes to the manuscript are indicated by track changes. A detailed point-by point response to each of the editor and reviewers’ point follows.

---

## [Decision Letter · Decision Letter 1]

19 Feb 2024

Tuberculin skin test and Interferon-gamma release assay agreement, and associated factors with latent tuberculosis infection, in medical and nursing students in Bandung, Indonesia

PONE-D-23-29807R1

Dear Dr. Apriani,

We’re pleased to inform you that your manuscript has been judged scientifically suitable for publication and will be formally accepted for publication once it meets all outstanding technical requirements.

Kind regards,

Mao-Shui Wang

Academic Editor

PLOS ONE

Additional Editor Comments (optional):

Reviewers' comments:

Reviewer's Responses to Questions

**Comments to the Author**

1. If the authors have adequately addressed your comments raised in a previous round of review and you feel that this manuscript is now acceptable for publication, you may indicate that here to bypass the “Comments to the Author” section, enter your conflict of interest statement in the “Confidential to Editor” section, and submit your "Accept" recommendation.

Reviewer #1: All comments have been addressed

2. Is the manuscript technically sound, and do the data support the conclusions?

Reviewer #1: Yes

3. Has the statistical analysis been performed appropriately and rigorously? 

Reviewer #1: Yes

4. Have the authors made all data underlying the findings in their manuscript fully available?

Reviewer #1: Yes

5. Is the manuscript presented in an intelligible fashion and written in standard English?

Reviewer #1: Yes

6. Review Comments to the Author

Reviewer #1: The authors have revised the manuscript according to the previous comments. I recommend the article to be accepted.

7. PLOS authors have the option to publish the peer review history of their article (what does this mean?). If published, this will include your full peer review and any attached files.

Reviewer #1: No

---

## [Editor Report · Acceptance letter]

7 Mar 2024

PONE-D-23-29807R1 

PLOS ONE

Dear Dr. Apriani, 

I'm pleased to inform you that your manuscript has been deemed suitable for publication in PLOS ONE. Congratulations! Your manuscript is now being handed over to our production team.

Kind regards, 

on behalf of

Dr. Mao-Shui Wang 

Academic Editor

PLOS ONE